# Multi-Angle Fast Neural Tangent Kernel Classifier

**Yuejing Zhai** [1],[†] **, Zhouzheng Li** [2],[†] **and Haizhong Liu** [1],[*]

1   School of Mathematics and Physics, Lanzhou Jiaotong University, Lanzhou 730070, China
2   School of Statistics, Southwestern University of Finance and Economics, Chengdu 611130, China
*   Correspondence: liuhzh@lzjtu.edu.cn
†   These authors contributed equally to this work and share first authorship.

**Abstract:** Multi-kernel learning methods are essential kernel learning methods. Still, the base kernel functions in most multi-kernel learning methods only with select kernel functions with shallow structures, which are weak for large-scale uneven data. We propose two types of acceleration models from a multidimensional perspective of the data: the neural tangent kernel (NTK)-based multi-kernel learning method is proposed, where the NTK kernel regressor is shown to be equivalent to an infinitely wide neural network predictor, and the NTK with deep structure is used as the base kernel function to enhance the learning ability of multi-kernel models; and a parallel computing kernel model based on data partitioning techniques. An RBF, POLY-based multi-kernel model is also proposed. All models use historical memory-based PSO (HMPSO) for efficient search of parameters within the model. Since NTK has a multi-layer structure and thus has a significant computational complexity, the use of a Monotone Disjunctive Kernel (MDK) to store and train Boolean features in binary achieves a 15–60% training time compression of NTK models in different datasets while obtaining a 1–25% accuracy improvement.

**Keywords:** neural tangent kernel; support vector acceleration; particle swarm algorithm; Boolean kernels; neural tangent kernel acceleration; multi-kernel learning; data splitting

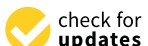



## 1. Introduction

Kernel methods, such as support vector machines (SVM) and kernel ridge regression, are an important class of machine learning methods. These methods implicitly map low-dimensional, indistinguishable data points to a high-dimensional space and teach linear learners in high-dimensional areas. This implicit mapping is generated by kernel function induction, so the choice of kernel function can often determine the performance of kernel methods.

When the sample features contain heterogeneous information, the sample size is large, the distribution of multidimensional data is irregular (e.g., NON-IID), or the data are not flat in the high-dimensional feature space, it is not reasonable to use a single simple kernel for mapping all samples. To better handle these kinds of data (such as building models on the data collected by personal terminal devices), vertical federated learning (VFL) is widely studied as a variant of federated learning (FL) [1,2]. The emergence of VFL stems from the concern for data privacy. It summarizes the parameters of each sub-model to train the global model (generally through neural networks). These sub-models are trained by vertical partition data with different attributes (i.e., feature dimensions) [3]. Based on similar ideas, there is also split learning [4]. Different from FL, it breaks the network structure. Each device retains only a part of the network structure, and the subnets of all devices form a complete network model. In the training process, different devices only perform forward or backward calculations on the local network and transmit the calculation results to the next device. Multiple devices complete the model training through the intermediate results of the joint network layer until the model converges [5]. "In network learning" is a particular FL structure [6], different from the classic FL. It runs distributed processing through the FL

training, reasoning and testing stages. The information exchanged by each node in various locations and the bandwidth requirements has also been thoroughly studied [7].

In contrast, using multi-core learning instead of single-core methods enhances the interpretability of the decision function, is more capable of handling large samples and variable data, and efficiently achieves better performance than single-core models [8,9]. Compared with a single kernel function, the multicore model has higher flexibility. The high-dimensional space after mapping multiple kernel functions is a combinatorial space made by combining various feature spaces, and the combinatorial space can combine different feature mapping capabilities of each subspace to map various features in the data by the most appropriate single kernel function, respectively, which more accurately represents the data in the new combinatorial space.

Multi-kernel learning methods have different classification methods according to different classification standards. According to the other construction methods and characteristics of multi-kernel functions, multi-kernel learning methods can be roughly divided into three categories: the composite kernel method combines multiple kernel functions with different factors [10]. It combines various kernel functions by directly summing or weighted summing the essential kernel functions. However, this method may lose some characteristic information of the original data; for all input samples, the weights corresponding to different cores are unchanged, and the samples are virtually averaged. Based on this consideration, a non-stationary combination method [11] is proposed, and each input sample is assigned with different weight coefficients. The multi-scale kernel [12] approach is relatively more flexible and can provide a more complete scale selection than the synthetic kernel method, which is more conducive to processing data with uneven distribution [13]. The above two methods are based on the premise of a linear combination of finite kernel functions, but they are not necessarily effective for some large-scale problems [14]. The expression ability of the decision function of multi-core fusion cannot be optimal everywhere. Compared with other methods, the infinite kernel method has a unique feature, that is, the number of basic kernels can be unlimited [15,16], and only these kernels need to be continuously parameterized [17].

Existing multicore model studies commonly use shallow structured kernel functions such as RBF and LINEAR, which have many shortcomings when dealing with large sample sizes [18], irregular data [19] or uneven data distribution [20], and to solve this problem, we propose a multicore learning method based on neural tangent kernel (NTK). Using NTK instead of the traditional shallow kernel function as the base kernel function for multicore learning, the optimal weight coefficients of each base kernel function and the optimal internal parameters of the kernel function model are iteratively derived using the improved particle swarm algorithm HMPSO, and the multicore decision function is derived and solved using a linear synthesis method. We also construct multi-kernel-RBF and multi-kernel-POLY models based on RBF and POLY using the same idea, conduct experiments on standard datasets and compare them with the well-known models.

In addition to the number of features aspect, constructing a model by changing the number of samples may be an effective method. This study discusses the NTK-SVM acceleration from the above two factors.

### 1.1. Our Contributions

- The traditional SVM is a single-kernel learning model, the SVM model with multi-kernel learning is introduced, using NTK, which is equivalent to an infinitely wide neural network, as the base kernel function for multi-kernel learning, and multi-kernel-NTK, RBF-Boolean, and POLY-Boolean models are selected for the classification task, where RBF- Boolean utilizes the multiscale multi-kernel idea, multi-kernel-NTK, POLY-Boolean utilizes the linear synthetic kernel idea.
- The HMPSO algorithm uses EDA to estimate and preserve the information of the historical optimal distribution of particles, which can avoid premature convergence of the model by competing among the information and maintaining fast convergence.

Its performance is better than most of the PSO improvement algorithms. We use the HMPSO algorithm to select various parameters in multi-kernel-NTK and other multi-kernel models.

- The single kernel SVM model has a limited search space, and the diagnostic results of the model are more sensitive and less robust to the kernel parameters and penalty parameters. The sensitivity of multi-kernel-NTK and NTK-SVM models to the penalty parameter C and network depth L is studied visually and comparatively.
- The NTK kernel has a multi-layer structure, which consumes a lot of time in its computation. Because of its special function construction (four-dimensional GRAM matrix), matrix decomposition and chunking acceleration techniques are difficult to be applied to the NTK kernel function. We extract the features belonging to the Boolean type from the original dataset and use the Boolean kernel (Monotone Disjunctive Kernel) to encode them into a binary square array for training, which improves the model accuracy and compresses the model training time at the same time.
- The Multi-NTK model is accelerated regarding the number of features and the multi-kernel model. We also used the data-cutting technique to construct the Block-NTK model to accelerate the model regarding the number of samples.

### 1.2. Article Structure

We review the contributions of related studies and where improvements can be made in Section 1, present recent research work in Section 2, introduce background techniques relevant to this study in Section 3, shows the experimental design details of the survey in Section 4, visualise and analyse the experimental results in Section 5, and discuss them in Section 6, and a conclusion in Section 7.

### 2. Related Works

Anirban et al. [21] proposed a multiple kernel learning embedded multi-objective swarm intelligence technique to identify the candidate biomarker genes from the transcriptomic profile of arsenicosis samples. Using multiple kernel learning MKL models, Chien et al. [22] assessed the predictive value of various clinical and MRI measures for disease activity. Jiang et al. [23] proposed a high-order norm-product regularized multiple kernel learning framework to optimize the discrimination performance. Stanton R et al. [24] explored different divergence measures on the values in the kernel matrices and reproducing kernel Hilbert space (RKHS). Fatemeh and Sattar [25] formulated multiple kernel learning in a bi-level learning paradigm consisting of the kernel combination weight learning (KWL) stage and the self-paced learning (SPL) stage. Yang et al. [26] focused on enhancing the original data representation by combining the gravitation-based method with a multiple empirical kernel approach. This paper proposes a sample-level method known as the gravitational balanced multiple kernel learning (GBMKL) method. Archibald et al. [27] developed a kernel learning backwards SDE filter method to estimate the state of a stochastic dynamical system based on its partial noisy observations. Tian et al. [28] proposed an efficient kernel method called reduced PSVM-2V (RPSVM-2V). It can provide a novel solution to process incomplete-view data and can also be adjusted to address large-scale complete-view learning problems efficiently. Saeedi [29] proposed an algorithm that uses recent advances in quantum sample-based Hamiltonian simulation to extend the existing quantum LS-SVM algorithm to handle the semi-supervised term in the loss. Guo et al. [30] proposed a novel semi-supervised multiple empirical kernel learning (SSMEKL) which enables various practical kernel learning to achieve better classification performance with a small number of labeled samples and many unlabeled samples.

Saeedi presented a quantum machine learning algorithm for training semi-supervised kernel support vector machines. Shi et al. [31] proposed a localized multiple kernel learning model with a nonlinear synthetic kernel (LMKL-D). A three-layer deep multiple kernel learning model trained the nonlinear synthetic kernel. Ding et al. [32] proposed a dynamic quantum particle swarm optimization algorithm (DQPSO). Bin et al. [33] used the artificial

neural network model to establish the objective functions for particle swarm optimization. The biogeography-based learning particle swarm optimization (BLPSO) is used to optimize the B-spline function parameters by Guo Qing et al. [34].

Polato et al. [35] proposed a new family of Boolean kernels for categorical data where features correspond to propositional formulas applied to the input variables. He also proposed an approach for extracting explanation rules from support vector machines [36]. The kernel idea is based on using kernels with feature spaces composed of logical propositions. Alfaro et al. [37] introduced a novel method for accelerated training of parallel support vector machines based on ensembles. Song et al. [38] proposed an accelerator for the SVM algorithm based on local geometrical information.

## 3. Background Techniques

### 3.1. Particle Swarm Algorithms

The particle swarm optimization (PSO) algorithm [39] is one of the evolutionary algorithms. It starts from a random solution and iterates to find the optimal solution. The solution quality is evaluated by fitness, but it is more straightforward than genetic algorithm rules, without the "crossover" and "variation" operations of genetic algorithms, and only follows the searched optimal values to find the global optimum.

PSO is initialized as a population of random particles (random solutions), where the position property of the particle is denoted as $x_i = (x_{i1}, x_{i2}, \cdots, x_{iD})$, and the direction of motion and distance of the particle is determined by the velocity: $v_i = (v_{i1}, v_{i2}, \cdots, v_{iD})$. In each iteration, the particle updates itself by tracking two extremes: the first one is the optimal solution found by the particle itself, called the individual extremum; the other extremum is the optimal solution found by the whole population, called the global extremum.

$$
\begin{aligned}
v_{id}^{k+1} &= w[v_{id}^k + c_1 r_1 (p_{ibest}^k - x_{id}^k) + c_2 r_2 (p_{gbest}^k - x_{id}^k)] \\
x_{id}^{k+1} &= x_{id}^k + v_{id}^{k+1}
\end{aligned}
\tag{1}
$$

where $i = 1, \cdots, N$ is the number of particles, $d = 1, \cdots, D$ is the dimension of the search space, $w$ is the inertia weight, $c_1, c_2$ is the acceleration factor, $r_1, r_2$ is the random number uniformly distributed on the interval [0,1], and $p_{ibest} = (p_{i1}, p_{i2}, \cdots, p_{iD})$ is the individual extremum, and $p_{gbest} = (p_{g1}, p_{g2}, \cdots, p_{gD})$ is the global extremum.

Li et al. [40] proposed an improved HMPSO algorithm. A distribution estimation algorithm is used to estimate and preserve the historical hoping $p_{best}$ information of the particles. Each particle has three candidate positions, generated from historical memories $H$, $p_{ibest}$ and $p_{gbest}$, respectively:

$$
\begin{aligned}
Pos_{i\_j}(t+1) &= (Pos_{i\_1}, Pos_{i\_2}, Pos_{i\_3}) \\
Pos_{i\_j}^d(t+1) &= x_{id}(t+1) + v_{id}(t+1) \\
x_{id}(t+1) &= Pos_{i\_mi}^d(t+1)
\end{aligned}
\tag{2}
$$

### 3.2. Multi-Kernel Learning

Compared with a single-kernel function, the multi-kernel model has higher flexibility. The high-dimensional space after the mapping of multiple kernel functions is a combined space made by combining various feature spaces, which can connect different feature mapping capabilities of subspaces and map different feature components of heterogeneous data by the most suitable single kernel function, respectively. The data can then be more accurately and reasonably expressed in the new space and improve the prediction accuracy of sample data.

The different construction methods and characteristics of multi-kernel functions can be divided into three categories: synthetic kernels, multiscale kernels, and infinite kernels. We replace the traditional kernel functions such as linear kernel, Gaussian kernel and polynomial kernel with NTK with deep structure as the base kernel functions of the multi-

kernel learning method, thus enhancing the representation capability of the multi-kernel learning method.

The linear combinatorial synthesis method is used to perform convex combinations of multiple basis kernel functions, and the final combined kernel function $K_{final}$ is expressed as:

$$K_{final} = \sum_{i=1}^{n} w_i K_i, \; w_i \geq 0, \; \sum_{i=1}^{n} w_i = 1, \tag{3}$$

where $K_i$ is the basis kernel function, $w_i$ is the weight coefficient of the basis kernel function, and $n$ is the number of basis kernel functions. Then, the decision function of the SVM-based multi-kernel learning method can be transformed into:

$$f(x) = \text{sgn}(\sum_{j=1}^{o} a_j y_j K(x_i, x) + b) \Rightarrow f(x) = \text{sgn}(\sum_{j=1}^{o} a_j y_j \sum_{i=1}^{n} w_i K_i(x_i, x) + b) \tag{4}$$

where $0 \leq a_j \leq C$ is a Lagrangian multiplier.

The linear combinatorial multi-kernel model has no basis for parameter selection and combination, which cannot satisfactorily solve the uneven distribution of samples and limits the representation capability of the decision function. In contrast, multi-scale kernel fusion is more flexible and provides a more complete choice of scales.

Among the widely used kernel functions, the Gaussian radial basis kernel is a typical multiscale kernel with generalized approximation capability. Take this kernel as an example and multiscale it (assuming it has translation invariance):

$$\begin{aligned} K(x,z) &= \exp(-\frac{\|x-z\|^2}{2\sigma^2}) \\ K_i(x,z) &= [\exp(-\frac{\|x-z\|^2}{2\sigma_1^2}), \cdots, \exp(-\frac{\|x-z\|^2}{2\sigma_m^2})] \end{aligned} \tag{5}$$

where $\sigma_1 < \cdots < \sigma_m$, when $\sigma$ is small, the model is prone to classify the drastically varying samples. When $\sigma$ is large, it can classify those samples that vary gently, resulting in a better generalization capability. The decision function of the SVM-based multi-kernel learning method can be transformed into:

$$\begin{aligned} f(x) &= f_1(x) + f_2(x) + \cdots + f_m(x) \\ f_1(x) &= \sum_{i=1}^{N} a_i K_1(x_i, x) + b_1 \\ &\vdots \\ f_m(x) &= \sum_{i=1}^{N} a_i K_m(x_i, x) + b_m \end{aligned} \tag{6}$$

### 3.3. Kernel Functions

#### 3.3.1. Boolean Kernel

Plato proposed a new family of Boolean kernel functions for categorical data, where the features correspond to propositional formulas applied to the input variables.

We use the Monotone Disjunctive Kernel in our study. For two Boolean variables $x, z \in \{0, 1\}$, An active disjunction of $d$ literals for $X$ can be defined as a set of $d$ elements taken from the space $U$. Anytime $\exists a, b \in U_d | a \in X \wedge b \in Z$, then $U_d$ is an active subset for $X$ and $Z$, so the value of the kernel is the number of active subsets $U_d$ in common between $X$ and $Z$. The definitions are as follows:

$$\begin{aligned} K_{MDK}^d(x,z) &= \binom{|U|}{d} - \binom{|U \backslash X|}{d} - \binom{|U \backslash Z|}{d} + \binom{|U \backslash (X \cup Z)|}{d} \\ &= \binom{n}{d} - \binom{n - \langle x, x \rangle}{d} - \binom{n - \langle z, z \rangle}{d} + \binom{n - \langle x, x \rangle - \langle z, z \rangle}{d} \end{aligned} \tag{7}$$

### 3.3.2. Neural Tangent Kernel

Jacot et al. [41], building on a series of studies such as Neal et al. [42] and Matthews et al. [43], first connected a neural network with all layers trained, infinitely large, and trained using gradient descent to a kernel predictor, expressing the dynamics of the neural network in terms of a representation through an ordinary differential equation, so that the training process of the DNN can be approximated with the help of NTK.

Specifically, let $x \in R^d$ be the input, $t^{(0)}(x) = x$ denotes input variables that are not nonlinearized, and $d_0 = d$ represents the width of the initial layer. Then, a neural network with $L$ hidden layers can be defined as follows:

$$
\begin{aligned}
f(w, x) \quad &= f^{(L+1)}(x) = W^{(L+1)} t^{(L)}(x) \\
&= W^{(L+1)} \sqrt{\tfrac{c_\sigma}{d_L}} \sigma(W^{(L)} \cdots \sqrt{\tfrac{c_\sigma}{d_1}} \sigma(W^{(1)}(x)))
\end{aligned}
\tag{8}
$$

$W^{(h)} \in R^{d_h \times d_{h-1}}$ is the weight matrix of layer $h$. The set of network parameters is $w = (W^{(1)}, \cdots, W^{(L+1)})$. All elements are initialized with independent standard normal distributions. $\sigma : R \to R$ is the activation function, RELU is recommended. $c_\sigma$ is the scaling factor, usually defined as 2, which can be used to avoid gradient explosion or disappearance in deep networks.

Jacot deduces that the dynamics of the neural network exhibit limiting behavior along the gradient trajectory when the width $d_1, d_2, \cdots, d_L \to \infty$ of each layer. Let $x, x' \in R^d$ be two data points, then the covariance kernel $\sum^{(h)}(x, x') = f^{(h)}(x) \cdot f^{(h)}(x')$ of the output of layer $h$ can be recursively defined as:

$$
\begin{aligned}
\Sigma^{(0)}(x, x') &= f^{(0)}(x) \cdot f^{(0)}(x') = x^T x', \\
\Lambda^{(h)}(x, x') &= \begin{pmatrix} \Sigma^{(h-1)}(x, x) & \Sigma^{(h-1)}(x, x') \\ \Sigma^{(h-1)}(x', x) & \Sigma^{(h-1)}(x', x') \end{pmatrix}, h \in [L] \\
\Sigma^{(h)}(x, x') &= c_\sigma E_{(u,v) \sim N(0, \Lambda^{(h)})}[\sigma(u)\sigma(v)],
\end{aligned}
\tag{9}
$$

It is important to note that this recursive form holds during initialization and training (gradient descent training of $\eta \to 0$ is required). Formally, NTK is shown to be the kernel in the limit case of:

$$
\begin{aligned}
\Theta(x, x') \quad &\triangleq \left\langle \frac{\partial f(w, x)}{\partial w}, \frac{\partial f(w, x')}{\partial w} \right\rangle \\
&= \sum_{h=1}^{L+1} \left\langle \frac{\partial f(w, x)}{\partial W^{(h)}}, \frac{\partial f(w, x')}{\partial W^{(h)}} \right\rangle
\end{aligned}
\tag{10}
$$

For the loss function $L$ of the network:

$$
\begin{aligned}
\frac{\partial \theta}{\partial t} \quad &= -\nabla_\theta L = -\nabla_\theta f(\theta, X)^T \nabla_{f(\theta, x)} L \\
\frac{\partial f(\theta, x)}{\partial t} \quad &= \nabla_\theta f(\theta, x) \frac{\partial \theta}{\partial t} \\
&= -\nabla_\theta f(\theta, x)^T \nabla_\theta f(\theta, x) \nabla_{f(\theta, x)} L \\
&= -\Theta(x, X) \nabla_{f(\theta, x)} L
\end{aligned}
\tag{11}
$$

By transforming the expression of the neural network at $t = 0$, through operations such as Taylor decomposition and differentiation, it can be shown that, while $\Theta(x, X)$ evolves in time to $\Theta_t(x, X)$, $\Theta_t(x, X)$ is shown to converge to a definite $\Theta_0(X, X)$ [43] and does not evolve with the training process, i.e., $\Theta_t(X, X) = \Theta_0(X, X)$. Thus, this differential equation can be expressed as:

$$
\frac{\partial f(\theta, X)}{\partial t} = -\Theta_0(X, X) \nabla_{f(\theta, X)} L.
\tag{12}
$$

## 4. Experimental Setup

### 4.1. Data Setup and Parameter Selection

In this paper, we use more than twenty UCI-labeled datasets to validate the performance of the proposed algorithm, which includes data from various fields such as sports, economy, medicine, and environment, etc. The performance of the multi-kernel-NTK multi-kernel model is validated on more than 10,000 pieces of data.

The HMPSO algorithm uses EDA to estimate and preserve the information of the historical optimal distribution of particles, which have their own advantages and thus can complement each other to avoid premature convergence and maintain fast convergence. Therefore, in this paper, we use the HMPSO algorithm to measure the basis kernel function, and calculate the weight parameters corresponding to the basis kernel function and the built-in parameters of the basis kernel function. We set the range of kernel function parameters as in Table 1.

**Table 1.** Support vector model parameter settings.

| Kernel | Para.1 | Range | Para.2 | Range | Para.3 | Range |
|--------|--------|-------|--------|-------|--------|-------|
| Poly | C | $0.001, \ldots, 1024$ | Coef0 | $0.02, \ldots, 2$ | degree | $1, 2, 3$ |
| Rbf | C | $0.001, \ldots, 1024$ | gamma | $0.1, \ldots, 50$ | | |
| MDK | C | $0.004, \ldots, 1024$ | d | $0.003, \ldots, 32$ | | |
| NTK | C | $0.004, \ldots, 128$ | dep | $1, \ldots, 13$ | fix | $0, \ldots, 12$ |

It should be noted that the above parameters are not the parameter settings obeyed by a single experiment, but the concatenation of the parameters taken by all our experiments, for example, we set the maximum width to 5 when studying the accuracy of the multi-kernel-NTK model, and the maximum width to 13 when studying its robustness.

### 4.2. Multi-Kernel-NTK and Other Multi-Kernel Models

We use the HMPSO algorithm to select various parameters in the multi-kernel-NTK model. The particle length in the HMPSO algorithm is set to 40, where the first 20 particles constitute the penalty parameter C of the support vector model, taking values in the range $2^{-8} \leq C \leq 2^{10}$. The remaining particles are the kernel function selection particles, denoted as $[r_1, r_2, \cdots, r_{20}]$, where $r_i$ takes values of 1 or 0, representing whether the corresponding kernel is selected or not, respectively.

The features of Boolean type in the original dataset are extracted and trained using the Monotone Disjunctive Kernel. At the same time, the data are saved in binary to improve the computational efficiency further. The NTK kernel function is used to construct a four-dimensional GRAM matrix based on the remaining features, and the two are combined and brought into the SVM training.

For the NTK-Boolean and POLY-Boolean models, we choose linear combinatorial synthetic multi-kernel models, while for the RBF-Boolean model, we choose 20 alternative basis kernels, which are constructed by a multiscale multi-kernel learning method, with the kernel parameters obtained by exponential growth, taking values. The range is $\sigma_i = 2^i \sigma, 1 = 0, 1, 2, \cdots$

We compare our proposed model with well-known models such as RF and Adaboost and perform statistical hypothesis testing and present the results in the "Discussion" section (Section 6). In the support vector machine training using NTK, a two-dimensional symmetric matrix is reconstructed from a four-dimensional tensor using the index as a Gram matrix in a general sense and involved in the computation, and the computational flow of our proposed primary model: NTK-Boolean model is shown in Figure 1.

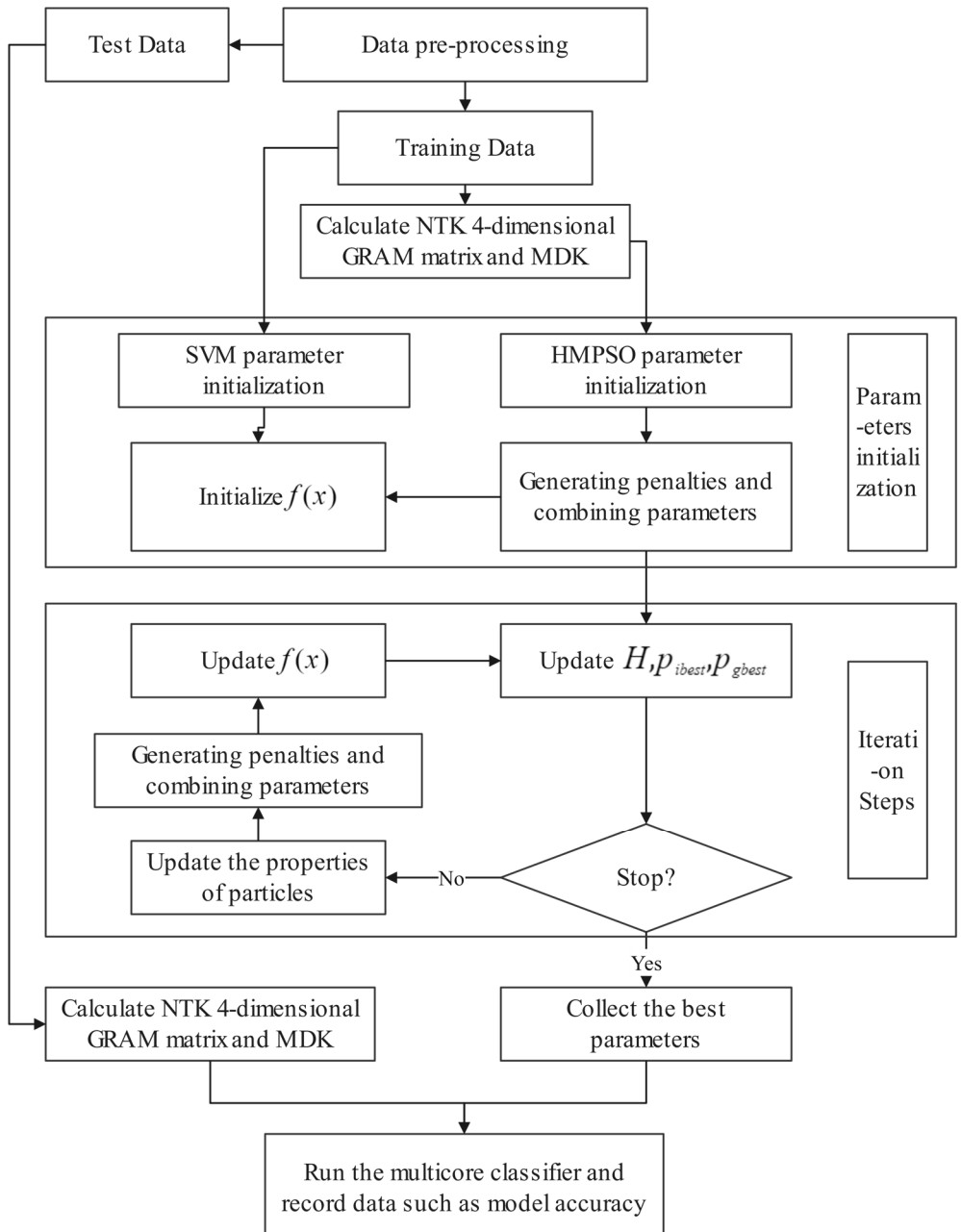

**Figure 1.** Flow chart of multi-kernel-NTK mode.

Relevant experiments were designed and completed to visualize the experimental results and discuss the accuracy ranking among the models and the acceleration achieved by our approach.

### 4.3. Multi-Kernel-NTK Model Robustness Test

We constructed the Block-NTK model from the perspective of sample size, using the idea of data cutting, dividing the original data into several subsets of the same size, and ensuring that each part of the subset has data of all categories, training on each subset in parallel using the NTK classifier, and aggregating the training results of different subsets, the time relationship between the NTK-SVM model processing different size datasets is not linear, so processing small datasets in parallel can theoretically obtain a minor time loss in processing large datasets.

*4.4. Multi-Kernel-NTK Model Robustness Test*

We discuss the robustness of the NTK model by referring to Arora et al. [44], Simon et al. [45], Novak et al. [46], and other studies that construct the NTK kernel function, following the setting of Arora et al. The computation of the NTK involves the construction of a four-dimensional tensor $K \in R^{L \times L' \times n \times n}$, where $n$ is the input sample size, the structure of the NTK kernel differs from the various existing kernel functions, and $L, L'$ is the hyperparameters specific to the NTK kernel. We have learned that the NTK is introduced from a neural network satisfying certain conditions and $L, L'$ represents the depth of that neural network and the depth at which the parameters are fixed, respectively (the maximum value is the total number of layers–1), so whether the structure of the network has an impact on the performance of the kernel function is an issue that must be considered, and we have also investigated the sensitivity of the model to the penalty parameter C. We visualized the experimental results and performed a comparative analysis.

## 5. Analysis of Experimental Results

*5.1. Comparison of Multi-Model Results*

Based on the multi-kernel-NTK model established above, we substituted the collected UCI dataset for the analysis; the selection of parameters in training follows the settings in Section 4.1. Other details follow the settings in Section 4.2. Using 70% of the dataset as the training set, we divided each original dataset 20 times and ensured that each training set and test set contained all categories. We summarized some of the training data in Table 2 and performed a statistical analysis. The optimal results for different datasets have been blacked out.

**Table 2.** Comparison of model results for selected datasets.

| Datasets | POLY | M-POLY | RBF | M-RBF | NTK | M-NTK | B-NTK |
|---|---|---|---|---|---|---|---|
| breast-cancer | 0.808 | 0.857 | 0.788 | 0.808 | 0.818 | 0.858 | 0.831 |
| credit-approval | 0.884 | 0.892 | 0.884 | 0.872 | 0.883 | 0.901 | 0.825 |
| echocardiogram | 0.892 | 0.881 | 0.862 | 0.896 | 0.854 | 0.898 | 0.942 |
| fertility | 0.9 | 0.9 | 0.9 | 0.91 | 0.9 | 0.92 | 0.875 |
| hepatitis | 0.805 | 0.799 | 0.844 | 0.822 | 0.818 | 0.882 | 0.856 |
| libras | 0.772 | 0.77 | 0.689 | 0.702 | 0.817 | 0.846 | 0.775 |
| Parkinsons | 0.918 | 0.873 | 0.938 | 0.883 | 0.938 | 0.968 | 0.877 |
| bridges-MATERIAL | 0.868 | 0.888 | 0.887 | 0.898 | 0.887 | 0.887 | 0.928 |
| bridges-REL-L | 0.784 | 0.845 | 0.824 | 0.875 | 0.765 | 0.875 | 0.755 |
| bridges-SPAN | 0.696 | 0.685 | 0.674 | 0.695 | 0.739 | 0.803 | 0.799 |
| bridges-T-OR-D | 0.902 | 0.943 | 0.902 | 0.943 | 0.887 | 0.907 | 0.926 |
| bridges-TYPE | 0.654 | 0.697 | 0.654 | 0.728 | 0.716 | 0.728 | 0.738 |
| german-credit | 0.76 | 0.772 | 0.75 | 0.798 | 0.786 | 0.798 | 0.770 |
| trains | 0.6 | 0.6 | 0.6 | 0.6 | 0.6 | 0.8 | 0.6 |
| wine | 0.989 | 0.953 | 0.978 | 0.908 | 0.978 | 0.989 | 0.870 |
| zoo | 0.96 | 0.96 | 0.96 | 0.9 | 0.9 | 0.96 | 0.8 |
| Friedman test | | | $H_0 : e_1 = e_2 = e_3 = e_4 = e_5 = e_6 = e_7$ $p = 0.000(reject\ H_0)$ | | | | |
| Kendall test | | | $H_0 : e_1 = e_2 = e_3 = e_4 = e_5 = e_6 = e_7$ $p = 0.000(reject\ H_0)$ | | | | |

In these experiments, we used different parameters for each model. For the POLY-based model, C was selected from $2^{-10}$ to $2^{10}$, coef0 was selected from 0.02 to 2, and degree was set at 1, 2, 3; for the RBF-based model, C was selected from $2^{-10}$ to $2^{10}$, gamma was chosen from 0.1 to 50; for NTK based model, C was selected from $2^{-8}$ to $2^7$, dep was selected from 1 to 6. Experimental results show that the multi-kernel-NTK support vector classifier model outperforms the RBF POLY support vector machine classifier, random forest, AdaBoost, and RBF-Boolean and POLY-Boolean models in more than 90% of different domain datasets and in datasets that do not reach optimal performance, the multi-kernel-NTK version is also very close to the optimal performance. Moreover, the accuracy of the

BLOCK model is maintained at a certain level, and the best accuracy is achieved in the three datasets. The statistical tests also demonstrate significant differences between the models.

### 5.2. Multi-Kernel-NTK Model Acceleration

To investigate the acceleration efficiency of the proposed model for the NTK regressor, we validated it on more than twenty UCI datasets, with the model parameters selected following the settings in Section 4.1 and other details following the grounds in Section 4.2, using 70% of the datasets as training sets. In comparing the temporal data, we normalize the running time of the original NTK model in different datasets with the running time of the multi-kernel-NTK model in each dataset as the origin, and the results of each experiment are shown in Figure 2 and Table 3.

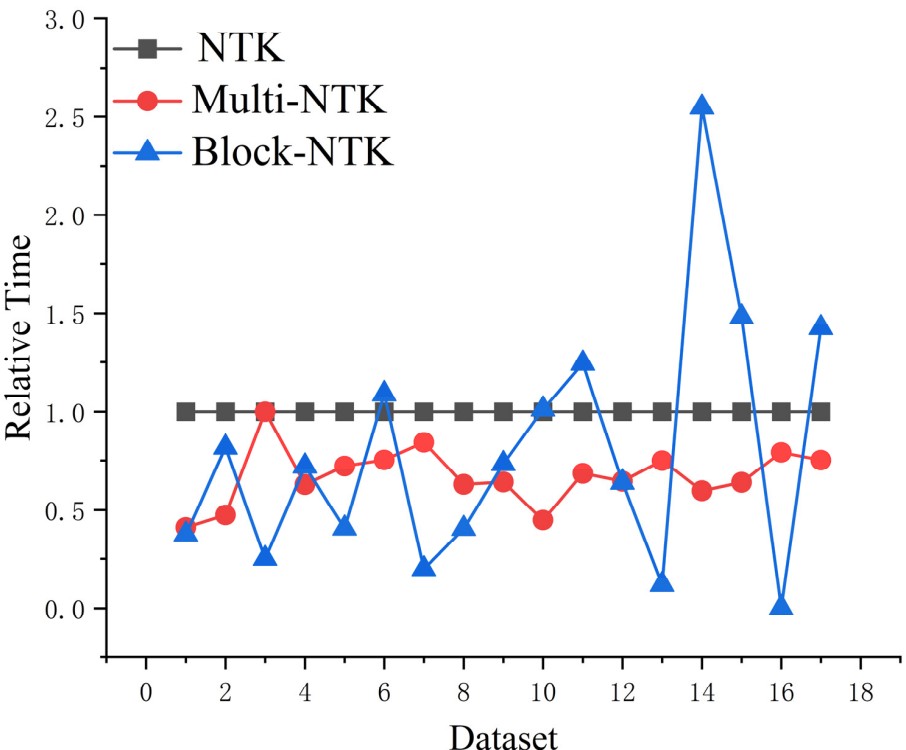

**Figure 2.** Model relative time comparison graph.

**Table 3.** Selection of Boolean variables for selected datasets and model efficiency.

| Dataset | Boolean(%) | M-Time Saving (%) | M-Acc Improv. (%) | B-Time Saving(%) | B-Acc Improv.(%) |
|---|---|---|---|---|---|
| breast-cancer 1 | 22.2 | 58.9 | 2.47 | 62.5 | 8.01 |
| breast-cancer 2 | 6.1 | 52.5 | 3.65 | 62.5 | 1.57 |
| credit-approval | 26.7 | 30.0 | 2.04 | 18.3 | −7.89 |
| echocardiogram | 30.0 | 37.2 | 5.15 | 74.9 | 10.30 |
| heart-cleveland | 23.1 | 27.9 | 8.46 | 28.0 | 15.15 |
| hepatitis | 63.2 | 24.7 | 7.83 | 59.3 | 4.62 |
| libras | 23.3 | 15.7 | 3.55 | −9.0 | −5.14 |
| Parkinsons | 13.6 | 37.1 | 3.20 | 80.2 | −5.96 |
| bridges-REL-L | 28.6 | 35.9 | 14.38 | 59.5 | −1.31 |
| bridges-T-OR-D | 28.6 | 55.1 | 2.25 | 26.6 | −9.92 |
| bridges-SPAN | 28.6 | 31.6 | 8.66 | −1.3 | 8.53 |
| bridges-TYPE | 28.6 | 35.5 | 22.15 | −24.4 | 3.07 |
| Australian-credit | 28.6 | 25.2 | 3.36 | 36.0 | 13.04 |
| zoo | 62.5 | 40.3 | 6.67 | 88.1 | −11.11 |
| lenses | 75.0 | 36.1 | 25.00 | 14.0 | 4.34 |
| waveform | 28.6 | 20.9 | 4.81 | −48.6 | 5.96 |
| wine | 30.8 | 24.9 | 1.12 | 99.8 | −0.80 |

For the NTK based model, C was selected from $2^{-8}$ to $2^7$, dep was selected from 1 to 6; for the Boolean kernel we used, C was selected from $2^{-8}$ to $2^{10}$, and d was selected from 0.003 to 32.

The second column in Table 3 shows the proportion of Boolean-type features extracted from the original dataset to the total number of parts, and the third column shows the percentage of time resources saved by the multi-kernel-NTK model compared with the original NTK classifier. The third column shows the improved classification accuracy of multi-kernel-NTK compared with the NTK model. The fourth column shows the percentage of time resources saved by the Block-NTK model compared with the original NTK classifier. The last column shows the improved classification accuracy of Block-NTK compared with the NTK model.

From Figure 2 and Table 3, it can be easily seen that our proposed multi-kernel-NTK model achieves higher computational efficiency in most datasets in different domains, saving up to nearly 60% of the time. At the same time, the model also improves the classification accuracy of the traditional NTK model, ranging from 1% to 25% in different datasets. For the BLOCK model, there may be a risk of deepening the time complexity in some datasets, but overall, the speedup is achieved in most datasets and the accuracy of the original model is relatively maintained.

### 5.3. Multi-Kernel-NTK Robustness Testing

#### 5.3.1. Network Parameters and Multi-Kernel-NTK

We tried to find the model factors that can affect the NTK model, as mentioned in Section 4.4, we investigated the effect of changing the SVM parameter C and NTK parameter L on the computational complexity of NTK separately. We selected several datasets from the UCI dataset and tested the robustness of NTK and multi-kernel-NTK models, respectively, at different network depths L, where the parameter L was taken from 1 to 12 and trained for all layers. The results are shown in Figure 3. Since the Block model also uses the NTK kernel function, we do not discuss its robustness.

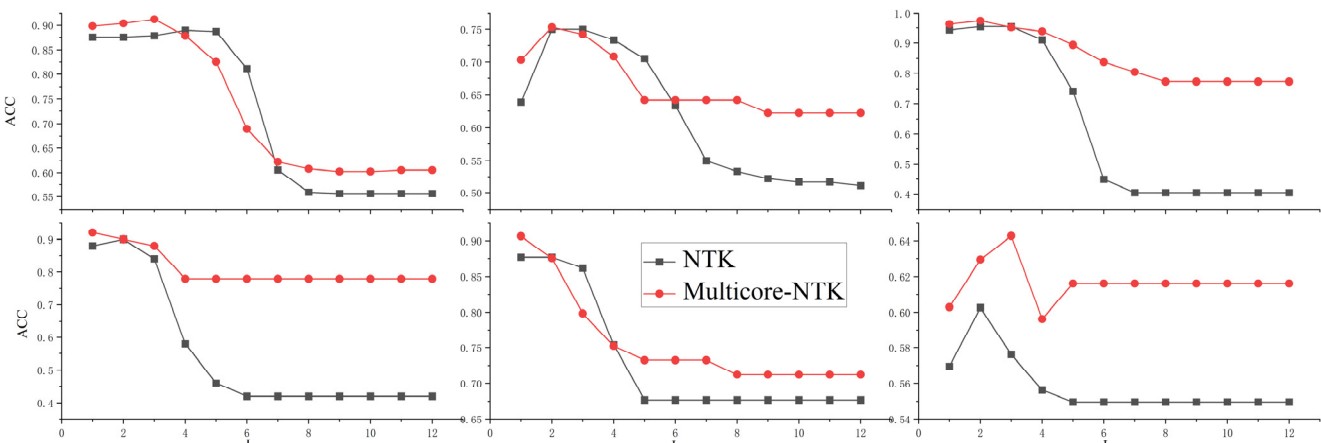

**Figure 3.** The effect of changing the network parameters L on the accuracy of the model, each graph corresponds to a different dataset.

Fifteen datasets are selected for display in Figure 3, where the red line represents the accuracy of the multi-kernel-NTK model as a function of L, and the black line represents the accuracy of the original NTK model as a function of L. An interesting trend of the NTK model can be found here, where the accuracy of the model tends to increase as the number of network layers increases, and when the number of layers is greater than five, as the number of network layers increases, the accuracy of the model. This is contrary to the classical notion of neural networks: more layers of the network tend to imply higher accuracy.

As can be seen from the figure, the multi-kernel-NTK model demonstrates better robustness as L increases, and even in some datasets the model accuracy gradually increases as the number of network layers increases.

### 5.3.2. Network Parameters and Multi-Kernel-NTK

As mentioned in Section 4.4, we investigated the effect of changing the SVM parameter C and NTK parameter L on the computational complexity of NTK, respectively, and selected several datasets from the UCI dataset and tested the robustness of NTK and multi-kernel-NTK models under different penalty parameters C, where the parameter C ranges from 0.004 to 128, and the results are shown in Figures 4 and 5.

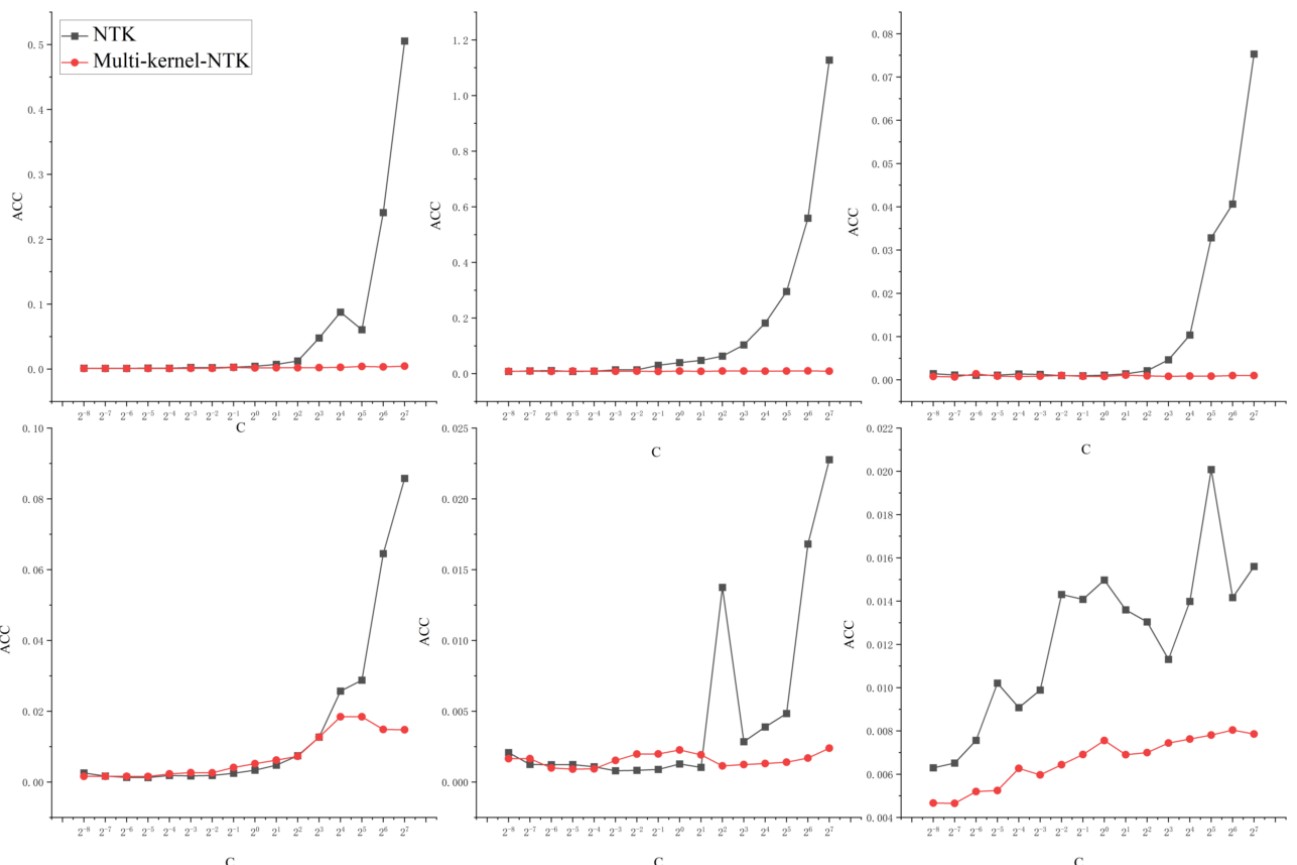

**Figure 4.** Effect of penalty parameter C on the training time of the model.

The red line represents the trend of the multi-kernel-NTK model with C, and the black line represents the trend of the original NTK model with C.

Another interesting trend of the NTK model can be found in Figure 4; the training time of the model increases exponentially as C increases, which is a property not possessed by other kernel functions. Still, our proposed multi-kernel-NTK model can mitigate this trend well and avoid the significant time expenditure that may result from an overly large parameter C. It can be seen from Figure 5 that as C increases, the multi-kernel model obtains better classification accuracy compared with the original model.

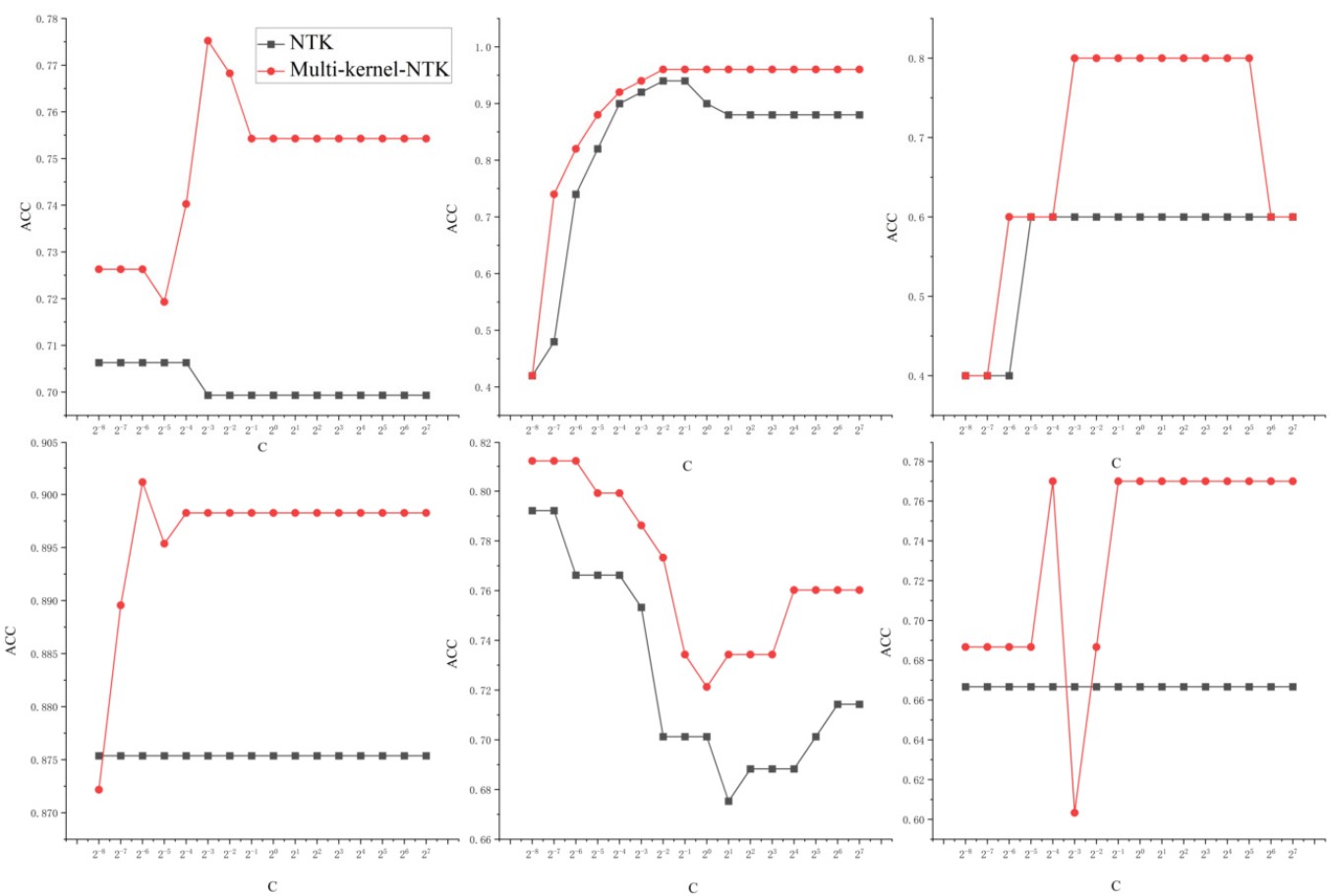

**Figure 5.** Effect of penalty parameter C on the accuracy of the model.

## 6. Discussion

The SVM model of multi-kernel learning is introduced, and NTK is used as the base kernel function of multi-kernel learning. More than twenty UCI datasets are selected, and a multi-kernel-NTK diagnostic model using HMPSO optimization is built for multi-domain classification tasks, which extracts features of sample data from multifaceted high-dimensional space and performs linear convex combination, which can better identify sample data and expand the search of the model space. At the same time, the multiscale multi-kernel learning method is used to construct 20 multiscale RBF kernel functions and MDK kernel functions to create the RBF-Boolean model and the POLY-Boolean model with a linear combination of POLY and MDK. The above models are compared with RF and Adaboost models, and the superiority of multi-kernel. The power of the NTK model is demonstrated by statistical tests. We constructed the Block-NTK model from the perspective of sample size, using the idea of data cutting, dividing the original data into several subsets of the same size, ensuring that each part of the subset has data of all categories, training on each subset in parallel using the NTK classifier, and aggregating the training results of different subsets.

As can be seen from Table 4, the accuracy of our proposed NTK improvement model outperforms the commonly used classification models in most multi-domain, different-size datasets. In Table 5, we show the relative time complexity of different models running different datasets and we mark the model with the shortest running time in each dataset as 1. The higher relative complexity of the model represents its higher time complexity. Table 5 shows that the decision tree model achieves the fastest speed in most datasets. Still, our proposed model is second only to the decision tree model in terms of average running rate while maintaining the best average classification accuracy.

**Table 4.** Acc-Comparison between our model and commonly used standard models.

| Datasets | RF | ADABOOST | KNN | Decision Trees | M-RBF | M-NTK | B-NTK |
|---|---|---|---|---|---|---|---|
| breast-cancer | 0.798 | 0.677 | 0.768 | 0.727 | 0.808 | 0.858 | 0.831 |
| credit-approval | 0.870 | 0.817 | 0.846 | 0.803 | 0.872 | 0.901 | 0.825 |
| echocardiogram | 0.846 | 0.831 | 0.815 | 0.8 | 0.896 | 0.898 | 0.942 |
| fertility | 0.9 | 0.86 | 0.880 | 0.9 | 0.91 | 0.92 | 0.875 |
| hepatitis | 0.818 | 0.792 | 0.831 | 0.701 | 0.822 | 0.882 | 0.856 |
| libras | 0.678 | 0.767 | 0.633 | 0.572 | 0.702 | 0.846 | 0.775 |
| Parkinsons | 0.876 | 0.856 | 0.876 | 0.845 | 0.883 | 0.968 | 0.877 |
| bridges-MATERIAL | 0.849 | 0.868 | 0.868 | 0.830 | 0.898 | 0.887 | 0.928 |
| bridges-REL-L | 0.765 | 0.706 | 0.667 | 0.706 | 0.875 | 0.875 | 0.755 |
| bridges-SPAN | 0.674 | 0.717 | 0.696 | 0.587 | 0.695 | 0.803 | 0.799 |
| bridges-T-OR-D | 0.843 | 0.882 | 0.902 | 0.804 | 0.943 | 0.907 | 0.926 |
| bridges-TYPE | 0.635 | 0.596 | 0.558 | 0.538 | 0.728 | 0.728 | 0.738 |
| German-credit | 0.75 | 0.73 | 0.724 | 0.696 | 0.798 | 0.798 | 0.770 |
| trains | 0.6 | 0.6 | 0.4 | 0.6 | 0.6 | 0.8 | 0.6 |
| wine | 0.966 | 0.955 | 0.966 | 0.876 | 0.908 | 0.989 | 0.870 |
| zoo | 0.96 | 0.8 | 0.92 | 0.94 | 0.9 | 0.96 | 0.8 |

**Table 5.** Time-Comparison between our model and commonly used standard models.

| Datasets | RF | ADABOOST | KNN | Decision Trees | M-NTK | B-NTK |
|---|---|---|---|---|---|---|
| breast-cancer | 124.19 | 16.05 | 7.41 | 1.63 | 1.13 | 1.00 |
| credit-approval | 112.83 | 14.69 | 9.74 | 1.00 | 1.32 | 16.32 |
| echocardiogram | 236.59 | 28.55 | 7.58 | 1.00 | 10.64 | 1.72 |
| fertility | 323.30 | 43.64 | 5.39 | 1.00 | 5.42 | 5.19 |
| hepatitis | 479.56 | 33.72 | 5.51 | 1.00 | 2.19 | 1.52 |
| libras | 279.17 | 46.18 | 9.43 | 7.13 | 1.00 | 3.97 |
| Parkinsons | 186.28 | 24.77 | 4.87 | 1.00 | 1.29 | 1.34 |
| bridges-MATERIAL | 302.00 | 39.42 | 5.05 | 1.00 | 1.38 | 1.36 |
| bridges-REL-L | 235.11 | 30.93 | 3.54 | 1.00 | 1.21 | 1.29 |
| bridges-SPAN | 221.42 | 31.12 | 3.79 | 1.00 | 2.19 | 1.39 |
| bridges-T-OR-D | 305.00 | 39.12 | 4.53 | 1.00 | 1.90 | 1.25 |
| bridges-TYPE | 306.06 | 38.88 | 4.70 | 1.00 | 3.29 | 2.02 |
| German-credit | 163.34 | 24.71 | 13.00 | 1.98 | 1.00 | 12.49 |
| trains | 122.63 | 19.45 | 17.42 | 1.53 | 1.12 | 1.00 |
| wine | 358.86 | 5.72 | 93.48 | 1.00 | 6.40 | 4.05 |
| zoo | 247.87 | 32.61 | 7.16 | 1.00 | 2.07 | 1.40 |

The acceleration performance and robustness of the multi-kernel-NTK model are investigated. It is found that compared with the traditional NTK model, the multi-kernel-NTK model achieves excellent acceleration performance in most datasets, saving up to nearly 60% of the running time and improving the model accuracy, ranging from 1% to 25% in different datasets.

Multi-kernel-NTK reduces the sensitivity of the deep NTK kernel to parameters and penalty parameters and improves the robustness of the model. When L is more significant than four, the NTK model accuracy gradually decreases as the number of network layers increases. At the same time, multi-kernel-NTK has better resistance to this phenomenon, and when $4 < L < 13$, multi-kernel-NTK performs better in all data. The average accuracy of multi-kernel-NTK in all datasets is improved by 21.6% compared with NTK. The training time of the NTK model increases exponentially with the increase in C. Multi-kernel-NTK model can alleviate this trend well; when $2 < C < 128$, the average time consumed by multi-kernel-NTK in all datasets is reduced by 93.6% compared with NTK. The average accuracy of multi-kernel-NTK in all datasets is improved by 8.3% compared with NTK when $0.04 < C < 128$.

## 7. Conclusions

Our main work is constructing accelerated NTK acceleration models in feature and sample size reduction, respectively. The experimental results on multi-domain datasets show that our proposed models achieve satisfactory acceleration results and significantly improve the model classification accuracy.

As seen in Table 3, the multi-kernel-NTK model achieves accuracy improvement and operational speedup in all datasets. Still, the Blocking-NTK model loses some accuracy and increases the computational complexity of the model in a small number of datasets. This may imply that the acceleration of the NTK model from the original data processing level is more suitable from the data feature size, and the computational complexity of the NTK classification model is not a simple linear relationship with the sample size of the original data.

Compared with other kernel classifiers, NTK undoubtedly has significantly substantial computational complexity. In addition to processing by datasets, the GRAM matrix of the NTK model can be optimized directly by decomposition techniques such as Kronecker product; however, how to effectively process the four-dimensional GRAM matrix of NTK by using tensor compression (decomposition) methods is a problem worth thinking about, and at the same time to ensure that the decomposed kernel matrix should satisfy certain conditions (e.g., the matrix is required to be semi-positive definite).

**Author Contributions:** Methodology, Y.Z.; Software, Y.Z.; Validation, Y.Z.; Investigation, Z.L.; Resources, Y.Z. and Z.L.; Data Curation, Y.Z. and Z.L.; Writing—Original Draft Preparation, Y.Z. and Z.L.; Writing—Review and Editing, Y.Z. and Z.L. and H.L.; Visualization, Y.Z.; Supervision, H.L.; Funding Acquisition, Y.Z. All authors have read and agreed to the published version of the manuscript.

**Funding:** This research was funded by Department of Education Research Project of Gansu Province (No. 2022cxzx-590).

**Institutional Review Board Statement:** Not applicable.

**Informed Consent Statement:** Not applicable.

**Data Availability Statement:** All UCI datasets used in this study are publicly available in: http://archive.ics.uci.edu/ml/index.php, accessed on 5 September 2022. The datasets generated during and/or analyzed during the current study are available from the corresponding author on reasonable request.

**Acknowledgments:** The successful completion of this study requires thanks to all participants who made outstanding contributions.

**Conflicts of Interest:** The authors declare no conflict of interest.

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
