# Peer review of "Multi-Angle Fast Neural Tangent Kernel Classifier"

_applsci, doi:10.3390/app122110876_

Round 1

Reviewer 1 Report

Essentially, this paper proposes a neural tangent kernel-based learning method; and shows that the regressor is equivalent to a wide neural network predictor. 

(i) How  \Theta_t(x,X) can be shown to converge to a definite \Theta_0(X,X) ?

(ii) More details are needed as to why/how Eq. (12) holds.

 (iii)  How the model accuracy evolves (i.e., increases) with the number of network size ?

(vi) The bibliography should be enriched: multi-kernel methods are mainly useful when there are multiple sets of data which are heterogenous or NON IID in distribution and each of them processed separately by one kernel; and the output models then combined or fused. In this context, a directly related work which studies this for SVMs is

M. Sefidgaran, R. Chor and A. Zaidi, "Rate-distortion bounds on the generalization error of distributed learning", The Thirty-Sixth Annual Conference on Neural Information Processing Systems (NeurIPS), 2022.

The authors should also refer and discuss, in the Introduction section, the strongly related line of work  which studies ways of merging the outputs of various kernels which are produced spatially at different location points, including (a) the so-called vertical federated learning of

] B. McMahan, E. Moore, D. Ramage, S. Hampson, and B. A. y Arcas, “Communication-efficient learning of deep networks from decentralized data,” in Proceedings of the 20th International Conference on Artificial Intelligence and Statistics, AISTATS 2017, vol. 54, 2017, pp. 1273–1282.

and

https://arxiv.org/abs/2202.04309

(b) the Split learning

O. Gupta and R. Raskar, “Distributed learning of deep neural network over multiple agents,” J. Netw. Comput. Appl., vol. 116, pp. 1–8, 2018. [Online]. Available: https://doi.org/10.1016/j.jnca. 2018.05.003 

and the in-network learning of

M. Moldoveanu and A. Zaidi, "On in-network leaning: a comparative study with Federated and Split Learning", 22nd IEEE International Workshop on Signal Processing Advances in Wireless Communications, Lucca, Italy, Sep. 2021.

Reviewer 2 Report

The authors proposed two types of acceleration models. They investigated deeply the acceleration performance and robustness of the proposed Multi-Kernel-Neural Tangent Kernel(NTK) model. In this investigation, they used very large datasets from various fields to validate the performance of the proposed algorithm.

They found very important results such as the average accuracy of Multi-kernel-NTK in all data sets is improved by 21.6% compared to NTK. Also, they found that the Multi-kernel-NTK model can alleviate training averaged time 93.6% compared with NTK.

I recommend to:

1)    Line 228: It is suggested to give same information in table, about:   Para.1 Range Para.2 Range Para.3 Range, Coef0, gamma, d, dep, C.

2)    Between line 309 and 316, there is a mistake in Table number (Table 3 instead Table 2)

3)    Standardize the references to the journal stander.

Reviewer 3 Report

The authors need to take in consideration the following suggestions:

1.     The introduction needs to be improved by discussing current articles because only a few new articles have been included in it.

2.     It is mandatory that the authors compare their results with other methodologies in order to validate or show that their contribution is important to the subject.

3.     In the section named “discussion”, I recommend presenting a table where the qualitative and quantitative features of your proposal and the other proposals reviewed in order to show the main advantages and disadvantages of your proposal.

4.     A new section named “Conclusions” needs to be added in order to separate it from the discussion section.

Round 2

Reviewer 1 Report

The authors have revised their manuscript in a rather satisfactory manner -- I recommend Accept.